# Local Variations in Carbohydrates and Matrix Lignin in Mechanically Graded Bamboo Culms

**DOI:** 10.3390/polym14010143

**Published:** 2021-12-31

**Authors:** Kexia Jin, Zhe Ling, Zhi Jin, Jianfeng Ma, Shumin Yang, Xinge Liu, Zehui Jiang

**Affiliations:** 1Key Lab of Bamboo and Rattan Science & Technology, International Center for Bamboo and Rattan, Beijing 100102, China; jinkexia@zju.edu.cn (K.J.); majf@icbr.ac.cn (J.M.); yangsm@icbr.ac.cn (S.Y.); 2Co-innovation Center of Efficient Processing and Utilization of Forest Resources, College of Chemical Engineering, Nanjing Forestry University, Nanjing 210037, China; jjling19@njfu.edu.cn; 3State Key Lab of Chemical Engineering, College of Chemical and Biological Engineering, Zhejiang University, Hangzhou 310027, China; 4Research Institute of Wood Industry, Chinese Academy of Forestry, Beijing 100091, China; lucy870826@163.com

**Keywords:** multilayered bamboo fiber, topochemistry, microscopic imaging, gradient micromechanics

## Abstract

The mechanical performance of bamboo is highly dependent on its structural arrangement and the properties of biomacromolecules within the cell wall. The relationship between carbohydrates topochemistry and gradient micromechanics of multilayered fiber along the diametric direction was visualized by combined microscopic techniques. Along the radius of bamboo culms, the concentration of xylan within the fiber sheath increased, while that of cellulose and lignin decreased gradually. At cellular level, although the consecutive broad layer (Bl) of fiber revealed a relatively uniform cellulose orientation and concentration, the outer Bl with higher lignification level has higher elastic modulus (19.59–20.31 GPa) than that of the inner Bl close to the lumen area (17.07–19.99 GPa). Comparatively, the cell corner displayed the highest lignification level, while its hardness and modulus were lower than that of fiber Bl, indicating the cellulose skeleton is the prerequisite of cell wall mechanics. The obtained cytological information is helpful to understand the origin of the anisotropic mechanical properties of bamboo.

## 1. Introduction

Bamboo, with the advantage of sustainability, extraordinary growth rate, ready availability, low weight, excellent mechanical strength and superior toughness, has been widely used as a structural material and bio-composite for industrial application [1,2]. Compared with other common building materials, bamboo is stronger than most timbers, and its strength-to-weight ratio is higher than that of common wood, cast iron, aluminum alloy and structural steel [3,4,5]. In terms of tissue types, the bamboo culm consists of fibers, parenchyma and conducting elements (including the xylem vessels, sieve tubes, and companion cells). As the source of the superior mechanical properties of bamboo, the fibers’ density and quality vary significantly along and across the bamboo culm. For example, along the diametric direction, the longitudinal tensile modulus of elasticity for the outermost layer was 3–4 times as high as that of the innermost layer, while the longitudinal tensile strength ranged from 115.94 to 328.15 MPa from the outermost layer to the innermost layer [6]. Numerous studies have focused on the mechanical properties of bamboo and bamboo composites [7,8,9]. However, the origin of the anisotropic mechanical properties across and along the culm are poorly understood.

Actually, much of the mechanical behavior of bamboo is governed by the properties of the cell wall, which, in turn, can be described in terms of the submicroscopic structure of the wall and the localization of cell wall components of cellulose, hemicelluloses, and lignin. Cellulose is the main structural fiber in the plant kingdom and has remarkable mechanical properties for a polymer: its Young’s modulus is roughly 130 GPa, and its tensile strength is close to and even more than 1 GPa [10]. The properties of hemicelluloses and lignin are similar to common engineering polymers. Lignin, for instance, has a modulus of roughly 3 GPa and a strength of about 50 MPa [11]. An important role of lignin in the wood cell wall is to function as a cross-linking matrix between moisture sensitive cellulose and hemicelluloses, thereby, lignin contributes to the mechanical rigidity [12]. The modulus of xylan varies from a value of 8 GPa at low moisture contents (0–10%) to 10 MPa at moisture contents near saturation (70%). Meanwhile, the incorporation of xylan in the cell wall, especially in the secondary wall, has a strength-enhancing effect on the joint strength individual fiber crossings [13]. The arrangement of the basic building blocks in bamboo cell walls and the variations in cellular structure give rise to a remarkably wide range of mechanical properties.

Unlike the typical three-layered structure of wood secondary wall, bamboo exhibits a polylamellate secondary wall with alternating broad and narrow lamellae that arise from the alternation in the orientation of cellulose microfibrils in a matrix of intertwined hemicelluloses and lignin. To date, macro- and micro-structural investigations have been reported comprehensively. For example, Huang et al. [14] and Wang et al. [15] have investigated the effect of different locations within the vascular bundle and non-multilayered fibers and the age of bamboo on mechanical properties. However, quite few works focus on the mechanical properties of multilayered fiber, to date. Since the multilayered fiber with alternating hierarchical structure was considered to be one of the factors that contribute to the high tensile strength of bamboo [16], insight of its structure, especially “seeing” bamboo macro- and micro-structure in a chemical sense, has great significance for full utilization of bamboo resources.

To the best of our knowledge, this work is the first report to reveal the fine mechanical details within the multilayered fiber of bamboo and correlate them with the chemical features of the cell wall. The in situ distribution of polysaccharides and lignin within the fiber located at bamboo green (Bg), bamboo timber (Bt) and bamboo yellow (By) were visualized by complementary microscopy techniques, including FT-IR microscopy and confocal Raman microscopy. The microscopes allowed correlative imaging of the same biomass sample under near-physiological conditions and at high chemical and spatial resolutions at the tissue, cellular, and molecular levels. In parallel, nanoindentation technique was applied to investigate the cell wall mechanical properties of fibers along the radius of bamboo culm. This information will contribute to greater fundamental understanding of the mechanical design of bamboo fiber cell walls and the mechanically graded structure of bamboo culms.

## 2. Experimental

### 2.1. Materials

A three-year-old moso bamboo (*Phyllostachys pubescens*) culm was collected from the local forest in Anhui Taiping experimental center, International Center for Bamboo and Rattan. Blocks of a length about 2–3 cm along the grain were cut from the middle part of the 10th internode (numbered from the ground level). Without any further preparation, a series of 15-μm-thick cross-sections, located at around the middle part of culm wall were cut from the freshly blocks on a rotary microtome (LEICA RM2165, Wetzlar, Germany) for the FT-IR and Raman imaging.

### 2.2. FT-IR Imaging Analysis

The FT-IR microspectroscopic imaging system (Spotlight400, PerkinElmer Ltd., Waltham, MA, USA), which combines a microscope with a FT-IR spectrometer, was used to obtain the FT-IR microspectroscopic spectra and image. Visible images were obtained using a charge-coupled device (CCD) camera, which made it possible to observe the specimen and select the area of interest for spectral analyses. FT-IR microspectroscopic image was obtained by a liquid nitrogen cooled mercury cadmium telluride (MCT) line array (16 × 1 element) detector. The air-dried bamboo section was placed over a KBr window supported on the Spotlight stage. All FT-IR images were taken using Spectrum IMAGE Software (Perkin Elmer) and collected in transmission mode in the region of 4000–750 cm^−1^ at 8 cm^−1^ spectral resolution with a 6.25 µm × 6.25 µm spatial resolution and eight scans co-added. The obtained IR spectra were then processed by using software Spectra IR developed by Perkin Elmer Inc. The functions of atmosphere correction, flat correction and baseline offset correction were applied in turn to create corrected spectra.

### 2.3. Confocal Raman Imaging Analysis

For Raman chemical imaging, the bamboo section was placed on a glass slide with a drop of distilled water, covered by a coverslip (0.17 mm thickness) and sealed with nail-polish to prevent evaporation during measurement. Raman spectra were acquired with a confocal Raman microscope (LabRam HR Evolution, Horiba Jobin Yvon, Paris, France). Measurements were conducted with an Olympus 100×Oil objective (PlanC N 100×, Oil, NA = 1.25) and a 532-nm laser. The Raman light was detected by an air-cooled, front-illuminated spectroscopic electron-multiplying charge-coupled device (EMCCD) behind a grating (300 grooves mm^−1^) spectrometer with a spectral resolution of 2 cm^−1^. For mapping, 0.3 μm steps were chosen and every pixel corresponds to one scan. The overview chemical images separated cell wall layers and marked the defined distinct cell wall areas to calculate the average spectra from the areas of interest. The Labspect6 software was used for spectral and image processing and analysis. Before a detailed analysis, the calculated average spectra were baseline corrected using the linear least squares algorithm.

### 2.4. Nanoindentation Test

A Hysitron TI 950 nanoindentation instrument (Triboindenter, Hysitron Inc., Minneapolis, MN, USA) was used to measure the nanoindentation modulus, nanoindentation hardness, yield strength, and creep of the materials by calculating the load-displacement curve derived from nano-indenter loading and unloading on the materials. The standard holding time was 50 s, while loading times or unloading times varied at 5 s, 15 s, and 25 s, respectively. A Berkovich diamond pyramid was used for indentation, with indentor tip radius of about 100 nm. Creep compliance tests were performed by using 300 μN, 400 μN, and 500 μN constant loads. Sample blocks for nanoindentation tests were vertically fixed on the cylindrical holder. The transverse (cross section) surface of the bamboo was polished by an ultra-microtome with a diamond knife to keep the cutting surface smooth. Before testing, the smoothed samples were conditioned for at least 24 h at 22 ℃ and 60% relative humidity in a room that housed the nano-indenter to ensure uniform moisture content. At least 35 valid points in cell corner (CC), outer layer of fiber secondary wall (SW-out) and inner layer of fiber secondary wall (SW-in) of each sample were tested, and the elastic modulus and hardness were obtained by averaging. All samples were treated equally to compare the mechanical properties on a relative basis.

## 3. Results and Discussion

### 3.1. FT-IR Chemical Image of Bamboo Culm

The FT-IR spectra of the bamboo fiber sheaths extracted from Bg, Bt and By are shown in Figure 1a. FT-IR spectra indicated peak changes in fingerprint regions at various tissues and cell locations. The main differences in the absorption spectra are visible at wavenumbers 1730, 1508, 1369, 1240, 1158, and 896 cm^−1^. The IR band at 1508 cm^−1^ corresponds to a C=C stretching the vibration of the aromatic rings of lignin. The carbohydrates peaks at 1730, 1369, and 1158 cm^−1^ are assigned, respectively, for unconjugated C=O in xylan, C-H deformation in cellulose and hemicelluloses, and C–O–C vibration in cellulose and hemicelluloses [17,18]. Moreover, the band at 1240 and 896 cm^−1^ is a diagnostic peak for cellulose by the C–O–C vibration and C–H deformation in cellulose, respectively [19,20]. The relative absorbance of characteristic FT-IR absorbance bands is plotted as a function of the fiber sheaths (Figure 1b). It was noted that the bands assigned to cellulose (1240 and 896 cm^−1^), and to lignin (1508 cm^−1^) showed obvious decrease in the absorbance peaks from Bg to By for fiber sheath, while the xylan band at 1730 cm^−1^ displayed an increase in the corresponding regions. Notably, the intensity of hemicelluloses related band at 1369 and 1158 cm^−1^ kept constant, probably due to these two band areas containing partial contribution from cellulose which displayed a declining tendency in band intensity.

Previously, the investigation of carbohydrate and lignin in the bamboo cell wall mainly focused on the total value of lignin in bamboo tissues, comprising all tissues, such as fibers, vessels and parenchyma, rather than specifying the lignin and values for individual cell types [21,22,23]. Actually, the extent of compositional variation was found to be influenced by the age of the culm, in certain ages by the position of the vascular bundle, and most strikingly, by the proportion of fibers within the vascular bundle and surrounding parenchyma. The FT-IR microspectroscopic data can be displayed as chemical images at specific wavelengths at the tissue level with a spatial resolution near 6.25 µm (Transmission mode). Figure 2 shows the functional group chemical images of bamboo culm transverse sections. The red color represents high intensity and the blue color stands for little or no intensity. Thus, chemical images at bands near 1730, 1508, and 1240 cm^−1^ can show the relative concentrations of xylan, lignin, and cellulose, respectively. The chemical images indicated that the xylan (Figure 2a) and lignin (Figure 2b) mostly accumulated in the fiber sheath within the vascular bundles and appeared much less in the ground parenchymatic regions. This is to be expected, because the vascular bundles are believed to be the main mechanical support for the whole bamboo culm. Interestingly, for a single vascular bundle, there is also a heterogeneity in compositional distribution. High lignin and xylan concentration was visualized in the outermost part of the fiber sheath. Comparatively, cellulose showed a more homogeneous distribution pattern for the fiber sheaths within the vascular bundles (Figure 2c). It has been revealed that the average tensile modulus for the bamboo fiber is three times higher than that of the parenchyma [24]. Cellulose has been proved to be the main structural component and has remarkable mechanical properties. Thus, the higher concentration of cellulose in the fiber sheaths area may partly explain the superior modulus and strength properties compared with parenchyma cells.

### 3.2. Confocal Raman Chemical Image of Bamboo Fiber

To further explore the chemical constituent distribution of bamboo fiber at the cellular and sub-cellular level in situ, confocal Raman microscopy with high spatial resolution (<0.5 µm) was employed. The strong band at 2897 cm^−1^ was assigned to the stretching of the C–H and C–H_2_ groups of carbohydrates [25]. The spectral fingerprints for the characteristic bands of lignin were identified at 1598 and 1656 cm^−1^, attributed to the stretching vibrations of the aromatic ring and ring-conjugated C=C bonds in coniferyl alcohol units, respectively [26].

By integrating over the CH and CH_2_ stretching vibrations (2800 cm^−1^ to 2918 cm^−1^), high intensity and thus high carbohydrates concentration were observed especially in the secondary wall (SW) of fiber (Figure 3a–c). Due to the high spatial resolution, the alternating broad (Bl) and narrow layers (Nl) can be easily differentiated in the polylamellate secondary walls, with the Bl having higher carbohydrates concentration than the Nl. Raman image by calculating the band ranges from 1540 cm^−1^ to 1660 cm^−1^ displayed the heterogeneity in lignin distribution. As shown in Figure 3d–f, lignin concentration was the highest in cell corner (CC) and compound middle lamella (CML). Within the SW, the lignin concentration decreased from the outer layer to the cell lumen, with relative higher concentration in the lumen edge as well as the interface between adjacent layers of the multilayered fiber.

To obtain better chemical composition changes among the Bg, Bt and By, the average Raman spectra in CC, CML, Bl, and Nl were extracted, respectively (Figure 4). It was noted that the Bg displayed higher lignin Raman intensity (1598 and 1656 cm^−1^) followed by Bt and fewest in By especially in CC and CML. In fiber SW, the average lignin Raman intensity of Bl was higher than Nl indicating higher lignin concentration in Bl. Along the diametric direction, although the Nl showed a similar lignin Raman strength, the Bl of Bg showed a relative higher lignin intensity. Meanwhile, the carbohydrates’ Raman intensity (2897 cm^−1^) both in Bl and Nl of the Bg was higher than that of the Bt and By. This detailed examination is consistent with the data with respect to the chemical change at specific wavelengths at tissue level observed by FT-IR, as shown in Figure 1.

In plant tissues, lignin does not exist in an independent polymer, but it is associated with cellulose and hemicelluloses, forming complexes with them through physical admixture and covalent bonds [27]. Specifically, in herbaceous plants, hydroxycinnamic acids (HCA) are attached to lignin and hemicelluloses, via ester and ether bonds as bridges between them forming lignin/phenolics–carbohydrate complexes [28,29]. By integrating over the band regions from 1131–1190 cm^−1^, assigned to cinnamoyl ester bond in HCA [30], it was demonstrated that HCA mainly deposited within the CC and CML of the corresponding cells (Figure 3g–i), showing a similar distribution of lignin.

To better observe the variability and distribution of carbohydrates (2897 cm^−1^), lignin (1598 cm^−1^) and HCA (1172 cm^−1^) concentrations in various layers of fiber wall, a double cell wall line scan was carried out (Figure 5). As the scan moved from the left lumen-secondary wall interface to the right lumen-secondary wall interface, lignin and attached HCA concentration gradually increased, reached the maximum value in CML, and then declined to a low value in the SW region of the right cell wall (Figure 5b,c). The heterogeneity in lignin distribution at a cellular level has also been reported in a two-month old moso bamboo and other woody biomass [15,31,32]. In the case of carbohydrates, the CML area with the highest lignin intensity showed the lowest concentration of 5.5 k intensity units (Figure 5a). Similarly, the carbohydrates distribution along the adjacent cell wall displayed a gradual increase from CML to lumen. Characteristically the Bl and Nl of fiber wall has the alternating compositional distribution pattern, with low carbohydrates concentration in the Nl.

To visualize the variation in carbohydrates to lignin ratio, the concentration of carbohydrates along the segment points was divided by lignin concentration. As shown in Figure 5d, the ratio decreased from 0.25 in SW of fiber to 0.075 in CML. Within the fiber SW, the ratio varied periodically, with the higher values in the Bl. By comparison, for several locations in the spruce and *Conus alba* L. fiber SW, the ratio was constant and reflected the fact that at these locations the concentrations of both lignin and carbohydrates increased or declined simultaneously [33,34]. The variation in distribution pattern largely stems from the mechanical roles of fibers in different species. As known, the ultrastructure of most of the bamboo fibers is characterized by thick polylamellate SW. This lamellation consists of alternating Bl and Nl with differing chemical compositions, which leads to an extremely high tensile strength, as demonstrated in engineering constructions with bamboo culms [35]. Comparatively, in the cell walls of xylem fibers or tracheids of normal wood that shares lower tensile strength, the polylamellate construction with different chemical composition does not exist. Additionally, it has been generally accepted and largely cited in the literature that the deposition of HCA is highly correlated with lignin [36,37]. However, it was found that the ratio of HCA to carbohydrates displayed a regular pattern that increased consecutively from 0.2 in the SW-lumen interface to 2.4 in CML (Figure 5e), while the ratio of HCA to lignin along the multi-layered fiber wall varied significantly (Figure 5f), indicating less dependence between these two polymers. During cell wall formation, the incorporation of lignin within the polysaccharide cell wall framework is generally regarded as the final stage of the typical differentiating process (Donaldson 2001). Thus, the accompanied distribution pattern between HCA and carbohydrates demonstrated that HCA probably participated in the cell wall biosynthesis prior to the lignin.

### 3.3. The Variation in the Mechanical Properties of Fiber Cell Walls

As shown in Figure 6a–c, the mean value of indentation modulus and hardness of bamboo fiber within a single fiber wall, displayed an obvious difference. The outer layer of fiber SW in all the Bg, Bt and By areas has higher modulus (19.59–20.31 GPa) and hardness (428–445 MPa) than the inner layer of fiber close to the lumen area with modulus of 17.07–19.99 GPa and hardness of 410–440 MPa. Although previous studies suggested that the microfibril angle is negatively correlated with the elastic modulus of fiber cell wall [38,39], the Raman spectral analysis result revealed a relatively uniform cellulose orientation distribution in the multilayered fiber by extracting the cellulose orientation sensitive band around 1097 cm^−1^ (Figure 6d). This result is consistent with the earlier typical microfibril angle model of multilayer fiber by Parameswaran & Liese [16]. Thus, the variation in the modulus was mainly due to the variation in the cell wall lignification level and its composition, which has been confirmed by lignin Raman line scan (Figure 5b). Similarly, in the spruce tracheid wall, the observed difference in the modulus of elasticity between developing and fully lignified cell walls is due to the filling of spaces with lignin and an increase in the packing density of the cell wall during lignification [40].

Additionally, along the radius bamboo culm the modulus and hardness also had a gradient trend. The Bg fiber presented a relatively higher modulus and hardness than that located at Bt and By due to a higher lignification degree as supported by the FT-IR and Raman results. Although the Bt and By displayed a relatively lower modulus and hardness than that of Bg, their values are still superior to those of other lignocellulosic biomass as listed in Table 1. This is probably because bamboo has a special cell wall structure with alternating broad and narrow layers, smaller microfibril angle and higher density than wood [35,41,42]. Earlier work has investigated anatomical, physical and mechanical properties of different wood and bamboo material, and implies that most properties were mainly governed by the fibers [4,42,43,44,45,46]. Moreover, this micromechanics data in individual fiber agreed well with the previous research that the longitudinal tensile modulus of elasticity of bamboo for the outermost layer was 3–4 times as high as that for the innermost layer [6], probably due to the density variations in the fiber sheath, which acts like a plant muscle within the bamboo culm. Thus, when the single bamboo fiber with various modulus values are assembled together to form a fiber sheath, it contributes to the superior macro-mechanical properties in Bg, though translating the extraordinary mechanical properties of micro-scale individual fiber to the macroscale fiber sheath will inevitably face the fundamentally non-ideal stress transfer.

Although the CC regions displayed the highest lignification level, its modulus (10.52–12.32 GPa) and hardness (346–385 MPa) were lower than that of fiber SW, which was probably due to the trace amount of the cell wall skeleton substance deposited in this region. Actually, it has been stated that the impact of lignin content on mechanics seems to depend on the specific structural configuration of the plant cell wall, with high microfibril angles likely to be a requirement for a visible impact of lignin on stiffness [52,53]. Since only fiber walls along the culm of a specific internode was examined, the results may not generally be valid for the whole tissue or plant. However, the findings pointed out that besides the structural variety and complexity, the heterogeneity in composition distribution also contributed to the mechanically graded bamboo culms.

## 4. Conclusions

Combined microscopic techniques have been used to non-destructively investigate the compositional heterogeneity and variation in cell wall mechanics in moso bamboo. At the tissue level, the fiber sheath has a higher concentration of carbohydrates and lignin than the ground parenchyma cells from Bg to By. For the multilayer bamboo fiber, the Bl revealed a higher carbohydrates and lignin concentration than the Nl, and the outer Bl showed a higher lignification degree than the inner Bl, yet the consecutive Bl revealed a relatively uniform cellulose orientation and concentration. Furthermore, the Bg displayed the highest elastic modulus and hardness followed by Bt and then by By, and the elastic modulus and hardness decreased from the outer Bl to the inner Bl. Comparatively, the CC displayed the highest lignification level but the lowest hardness and modulus.

## Figures and Tables

**Figure 1 polymers-14-00143-f001:**
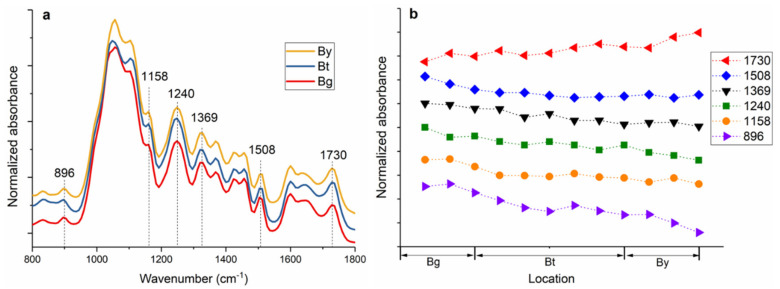
(**a**) FT-IR spectra of fiber sheath extracted from Bg to By; (**b**) Change in relative intensity of absorbance bands for fiber sheath from Bg to By.

**Figure 2 polymers-14-00143-f002:**
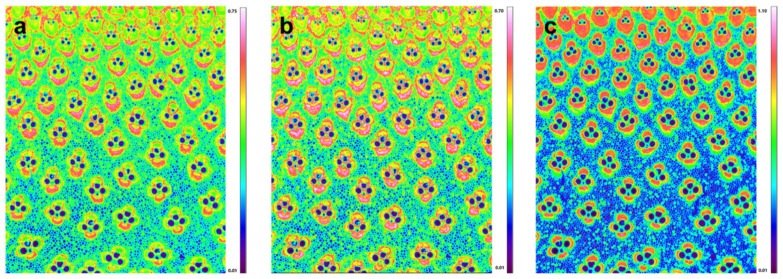
FT-IR images of the relative concentration and distribution of xylan (**a**), lignin (**b**), and cellulose (**c**).

**Figure 3 polymers-14-00143-f003:**
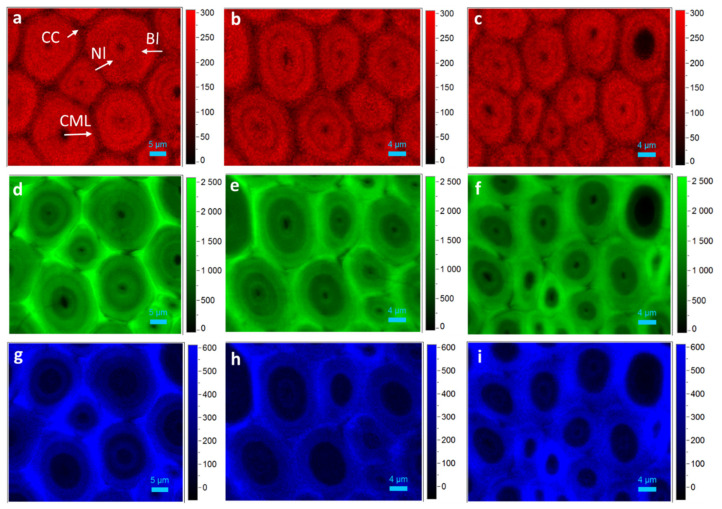
Raman images showing the distribution of fiber wall components from Bg (**a,d,g**), Bt (**b,e,h**) and By (**c,f,i**). (**a**–**c**) Carbohydrates, 2800–2918 cm^−1^; (**d**–**f**) Lignin, 1540–1660 cm^−1^; (**g**–**i**) HCA, 1120–1200 cm^−1^.

**Figure 4 polymers-14-00143-f004:**
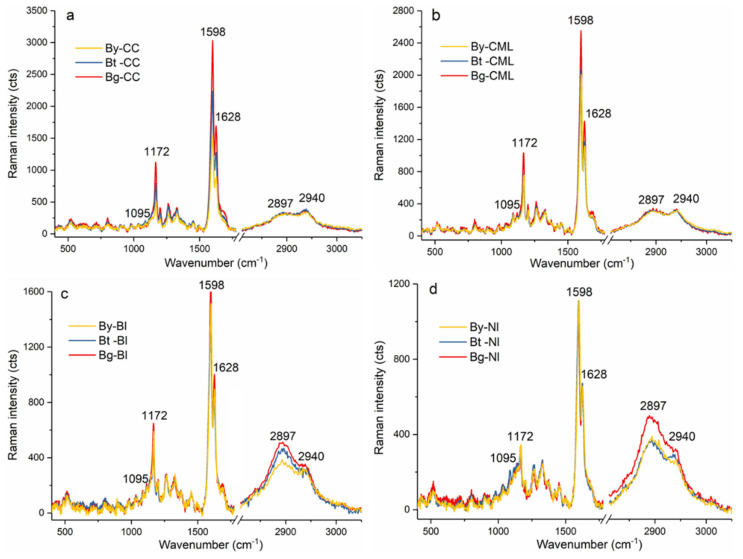
Average Raman spectra extracted from the CC (**a**), CML (**b**), Bl (**c**) and Nl (**d**) of fiber located at Bg, Bt and By.

**Figure 5 polymers-14-00143-f005:**
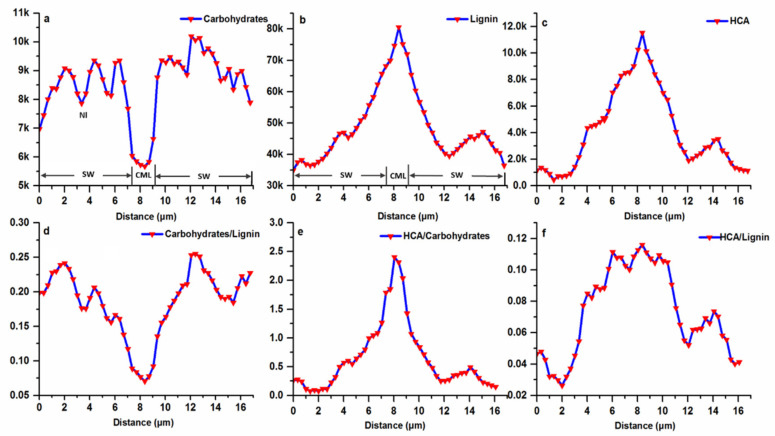
Raman line scan of Bg fiber wall, (**a**) Carbohydrates; (**b**) Lignin; (**c**) HCA; (**d**) Carbohydrates to lignin ratio; (**e**) HCA to carbohydrates ratio; (**f**) HCA to lignin ratio.

**Figure 6 polymers-14-00143-f006:**
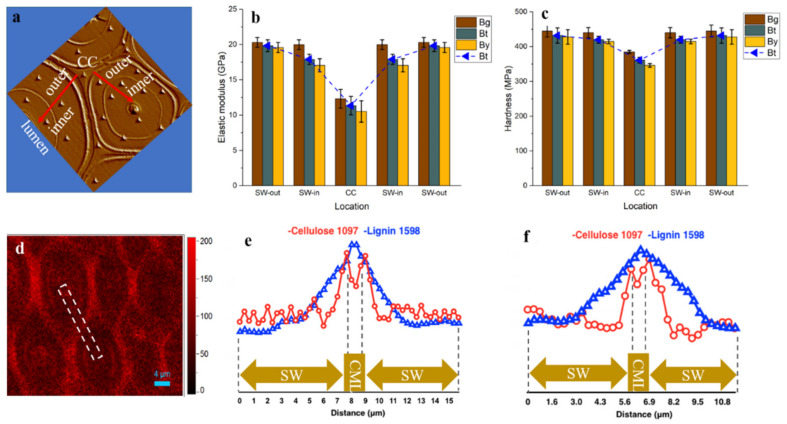
(**a**) The image obtained after indenting shows the actual position of indents; (**b,c**) The elastic modulus and hardness of bamboo fiber at different locations; (**d**) Raman image showing the cellulose microfibrils orientation distribution by integrating over 1097 cm^−1^; (**e,f**) the Raman line scan of Bg (**e**) and Bt (**f**) fiber wall. The white dotted line indicates the typical region of the Raman spectra collected. SW-out: outer layer of fiber secondary wall; SW-in: inner layer of fiber secondary wall.

**Table 1 polymers-14-00143-t001:** Fiber cell wall mechanical properties of different materials biomass.

Species	Modulus (GPa)	Hardness (GPa)	References
Spruce	17.1	0.38	[40]
Masson pine	19.18	0.53	[41]
Chinese fir	17.8	0.42	
Manchurian ash	17.5	0.48	[47]
Loblolly pine	14.2–17.6	0.43–0.53	[48]
Hemp	12.3	0.41	[49]
Cotton stalk	16.3	0.85	
Flax stalk	17.4	0.39	[50]
Bamboo (*Dendrocalamus farinosus*)	18.56	410.7	[51]
Bamboo (moso)	19.59–20.31	0.43–0.45	This work

## Data Availability

All data supporting the findings of this study are available within the article.

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
