# Peer review of "Local Variations in Carbohydrates and Matrix Lignin in Mechanically Graded Bamboo Culms"

_polymers, 2021, doi:10.3390/polym14010143_

Round 1
Reviewer 1 Report
„Local variations in carbohydrates and matrix lignin in mechanically graded bamboo culms” is an interesting paper that describes the mechanical properties of bamboo cell wall in relation to its chemical composition. I suggest adding more technical details and practical conclusions to the manuscript (see the comments in the pdf file attached). I recommend the paper for publishing after minor revision.

Author Response
We have modified the description in the revised manuscript. Bamboo is stronger than most of timbers, where the timbers means the common building materials, such as the Loblolly pine, Douglas-fir.

Reviewer 2 Report
The paper is interesting and touches on interesting issues in terms of bamboo biomass parameters and strength. In my opinion, the review lacks information on lignocellulosic biomass, which includes wood and bamboo (which is counted as wood in a way, despite its genetic identity and the fact that it belongs to grasses). In my opinion, a list of trees (various selected species and bamboo 'wood') should be included here. Pointing out similarities and differences. The paper lacks treatment of statistical material. The question is whether it is possible to perform e.g. an ANOVA test or whether the samples are individual/single? Below he presents examples of articles that can be used to further elaborate on the issues he points to.
https://doi.org/10.1007/s10086-017-1687-3
https://doi.org/10.5658/WOOD.2018.46.6.656
https://doi.org/10.3390/f12020223
https://doi.org/10.3390/en14102968
https://doi.org/10.1007/s00107-012-0619-6
Conclusions should take the form of maximally general sentences confirming or refuting the research hypothesis presented. At the same time, their formulation should be as general as possible (so that they can be related to other materials) and not focus on description and repetition of results or relations described in the results formulated in slightly different sentence forms.
Literature and citation should be corrected in the article according to the guidelines for the journal.
Author Response
- The paper is interesting and touches on interesting issues in terms of bamboo biomass parameters and strength. In my opinion, the review lacks information on lignocellulosic biomass, which includes wood and bamboo (which is counted as wood in a way, despite its genetic identity and the fact that it belongs to grasses). In my opinion, a list of trees (various selected species and bamboo 'wood') should be included here. Pointing out similarities and differences.
Replying: Thank you very much for your kind comments. The comparison of fiber cell wall mechanical properties of bamboo and other lignocellulosic biomass were listed in the revised manuscript.
- The paper lacks treatment of statistical material. The question is whether it is possible to perform e.g. an ANOVA test or whether the samples are individual/single? Below the presents examples of articles that can be used to further elaborate on the issues the points to: https://doi.org/10.1007/s10086-017-1687-3; https://www.bamboo-concept.com/wp-content/uploads/2020/12/Bamboo20Properties.pdf; https://doi.org/10.5658/WOOD.2018.46.6.656; https://doi.org/10.3390/f12020223; https://doi.org/10.3390/en14102968; https://doi.org/10.1007/s00107-012-0619-6
Replying: Thank you very much for your kind comments. Indeed, using ANOVA could well reveal the correlation between chemical composition and mechanical properties of the secondary wall. However, for FT-IR and Raman imaging a slice of 15-μm-thick cross-section was selected when transmission mode was set, while the nanoindentation test normally used tip polished bamboo strips. So, it is almost impossible to obtain the composition and mechanical information for the same cell and correlate these two sets of data directly.
- Conclusions should take the form of maximally general sentences confirming or refuting the research hypothesis presented. At the same time, their formulation should be as general as possible (so that they can be related to other materials) and not focus on description and repetition of results or relations described in the results formulated in slightly different sentence forms.
Replying: we have modified the description in the revised manuscript.
- Literature and citation should be corrected in the article according to the guidelines for the journal.
Replying: Literature and citation have been corrected in the revised manuscript.

Round 2
Reviewer 2 Report
Many thanks for making the suggested changes to the article. I still miss the inclusion in the literature review of information on the structural and chemical structure of grasses (bamboo) trees (deciduous and coniferous trees) and also e.g. palms. These features in a way explain the properties studied by the authors. Part of this discussion appeared when discussing the results, but in my opinion, the literature review is the place where this problem should be raised and highlighted.
Author Response
Thank you very much for your kind suggestion. Indeed, the mechanical behavior of different biomass are influenced by the structural and chemical structure. For cell wall mechanical properties investigated by Nanoindentation, the main factors here we need to consider are chemical composition and microfibril angles (MFA).
It has been reported that the longitudinal Young’s modulus of isolated cellulose, hemicelluloses and lignin in wood is 167.5 GPa, 7.0 GP and 2.0 GPa [1]. Obviously, the cellulose is responsible for the cell wall mechanical strength since its stiffness is more than 80 times that of lignin. Also, the hardness is increased with the lignin content since the lignin fills the space between the individual cellulose fibrils thus leading an increase of the packing density of the cell wall. Gindl et al. [2] has reported that the developing cell with average lignin content of 0.102 g/g had lower hardness (290 MPa) than that of the mature cell (380 MP) with lignin content of 0.209 g/g. Structurally, the effect of the MFA on cell wall modulus should not be neglected, since it was revealed that the cell wall modulus decreased with increasing MFA [3].
Similarly, the present work has shown that the outer of the secondary wall fiber with similar cellulose content and MFA but higher lignin content has higher modulus (19.59-20.31 GPa) and hardness (428-445 MPa) than the inner one (the corresponding values are 17.07-19.99 GPa and 410-440 MPa, respectively), indicating the increase of hardness is probably attributed to lignin content. This conclusion also applies to bamboo green and bamboo yellow at tissue level as listed in the Table 1. When compared with other biomass materials, such as Windmill palm, it was found that the lignin content was closely related with the cell wall hardness, while cell wall modulus was mainly determined by both the cellulose content and MFA.
However, quite a few literatures provide comprehensive information including the chemical composition, MFA and cell wall mechanical properties. For example, Wu et. al [4] investigated the MFA and cell wall mechanical properties of 10 hardwoods, while the chemical composition information were ignored. Although the chemical composition of these tree species can be found in other literature, the chemical composition of the same tree species vary with planting place, harvest time, sampling location (ring and height), etc. For example, the latewood of Loblolly pine contains 32.3% lignin and 34.8% cellulose at 1 m height, while at 10 m height the corresponding values are 28.9% and 39.3% [5]. Therefore, we think it is not rigorous enough to include the Table 1 in the manuscript. Moreover, unlike the in situ chemical imaging obtained by microscopic techniques at cellular and sub-cellular level, chemical composition information obtained by wet chemical analysis reflects ensemble average results of the whole plant. Therefore, it make more sense to tie the cell wall mechanical properties to the in situ chemical composition at cell wall in case of some details were overlooked. In this work, we first revealed that the outer broad layer of multilayered bamboo fiber has higher elastic modulus (19.59-20.31 GPa) than that of inner one (17.07-19.99 GPa). However, there are still many works about the cell wall mechanics of different biomass need further study in the future, such as the effect of hemicellulose content on the modulus and hardness.
Table 1 Structural and chemical structure of different biomass fiber cell wall
|
Natural fiber |
Cellulose (%) |
Lignin (%) |
MFA (⸰) |
Modulus (GPa) |
Hardness (GPa) |
References |
|
Spruce (developing cells) |
|
|
|
15.3 |
290 |
[2] |
|
Spruce (mature cells) |
|
|
|
17.1 |
380 |
|
|
49.4 |
27.7 |
20 |
|
|
[3, 6] |
|
|
Poplar |
49 |
23.2 |
|
|
|
[7] |
|
|
|
15.8 |
16.9 |
490 |
[4] |
|
|
Manchurian ash |
|
|
12 |
17.5 |
480 |
|
|
56.4 |
13.4 |
|
|
|
[8] |
|
|
Asian White Birch |
|
|
11.1 |
17.5 |
450 |
|
|
Loblolly pine |
43.1 |
28.1 |
|
|
|
[5] |
|
|
|
30 |
14.2 |
440 |
[3] |
|
|
|
|
15 |
16.3 |
410 |
||
|
Chinese fir |
|
|
15.6 |
17.8 |
420 |
[9] |
|
Masson pine |
|
|
12.6 |
19.18 |
530 |
|
|
49.2 |
28.4 |
|
|
|
[10] |
|
|
Windmill palm |
35.4 |
36.4 |
38.9 |
13.2 |
500 |
[11] |
|
Date plam |
45.1 |
16.9 |
26.1 |
14.3 |
470 |
[12] |
|
Moso bamboo |
56.4 |
22.6 |
|
18.4 |
450 |
[13] |
|
|
|
9.8 |
|
|
[9] |
|
|
Moso Bamboo (green) |
47.1 |
31.4 |
|
20.31 |
450 |
This work |
|
Moso Bamboo (yellow) |
47.0 |
27.5 |
|
19.59 |
430 |
References
- Bergander, A. and L. Salmen, Cell wall properties and their effects on the mechanical properties of fibers. Journal of Materials Science, 2002. 37(1): 151-156.
- Gindl, W., H.S. Gupta, and C. Grünwald, Lignification of spruce tracheid secondary cell walls related to longitudinal hardness and modulus of elasticity using nano-indentation. Canadian Journal of Botany, 2002. 80(10): 1029-1033.
- Tze, W.T.Y., et al., Nanoindentation of wood cell walls: Continuous stiffness and hardness measurements. Composites Part A: Applied Science and Manufacturing, 2007. 38(3): 945-953.
- Wu, Y., et al., Use of Nanoindentation and Silviscan to Determine the Mechanical Properties of 10 Hardwood Species. Wood and Fiber Science, 2009. 41(1): 64-73.
- Kretchmann, D.E., Cramer, S.M, The role of earlywood and latewood properties on dimensional stability of loblolly pine.Proceedings of the Compromised Wood Workshop, 2007 January 29-30. Christchurch, NZ: Wood Technology Research Centre, School of Forestry, University of Canterbury, 2007: pages 215-236.
- Butler, E., et al., Characterisation of spruce, salix, miscanthus and wheat straw for pyrolysis applications. Bioresour Technol, 2013. 131: 202-9.
- Sun, Q.N., et al., Effect of lignin content on changes occurring in poplar cellulose ultrastructure during dilute acid pretreatment. Biotechnology for Biofuels, 2014. 7.
- Guo, X.j., et al., Influence of extractives on mechanism of biomass pyrolysis. Journal of Fuel Chemistry and Technology, 2010. 38(1): 42-46.
- Huang, Y.H. and B.H. Fei, Comparison of the Mechanical Characteristics of Fibers and Cell Walls from Moso Bamboo and Wood. Bioresources, 2017. 12(4): 8230-8239.
- Wang, X., et al., Effects of thermal modification on the physical, chemical and micromechanical properties of Masson pine wood (Pinus massoniana Lamb.). Holzforschung, 2018. 72(12): 1063-1070.
- Li, J., et al., Structural, chemical, and multi-scale mechanical characterization of waste windmill palm fiber (Trachycarpus fortunei). Journal of Wood Science, 2020. 66(1).
- Bourmaud, A., et al., Exploring the potential of waste leaf sheath date palm fibres for composite reinforcement through a structural and mechanical analysis. Composites Part A: Applied Science and Manufacturing, 2017. 103: 292-303.
- Li, Y., et al., The effects of thermal treatment on the nanomechanical behavior of bamboo (Phyllostachys pubescens Mazel ex H. de Lehaie) cell walls observed by nanoindentation, XRD, and wet chemistry. Holzforschung, 2017. 71(2): 129-135.
